# Acute Ozone-Induced Transcriptional Changes in Markers of Oxidative Stress and Glucocorticoid Signaling in the Rat Hippocampus and Hypothalamus Are Sex-Specific

**DOI:** 10.3390/ijms24076404

**Published:** 2023-03-29

**Authors:** Matthew C. Valdez, Danielle L. Freeborn, Padmaja Vulimiri, Joseph M. Valdez, Urmila P. Kodavanti, Prasada Rao S. Kodavanti

**Affiliations:** 1Neurological and Endocrine Toxicology Branch, PHITD, CPHEA, ORD, US Environmental Protection Agency, Research Triangle Park, NC 27711, USA; 2Oak Ridge Institute for Science and Education Research Participation Program, U.S. Department of Energy, Oak Ridge, TN 37831, USA; 3Cardiopulmonary and Immunotoxicology Branch, PHITD, CPHEA, ORD, US Environmental Protection Agency, Research Triangle Park, NC 27711, USA

**Keywords:** air pollution, ozone, oxidative stress, bioenergetics, genomics, neurotoxicity, sex-specific effects, glucocorticoid signaling

## Abstract

Exposure to a prototypic air pollutant ozone (O_3_) has been associated with the activation of neuroendocrine stress response along with neural changes in oxidative stress (OS), inflammation, and Alzheimer’s disease-like pathologies in susceptible animal models. We hypothesized that neural oxidative and transcriptional changes induced by O_3_ in stress responsive regions are sex-dependent. Male and female adult Long–Evans rats were exposed to filtered air or O_3_ for two consecutive days (0.8 ppm, 4 h/day) and brain regions were flash-frozen. Activities of cerebellar OS parameters and mitochondrial complex I, II, and IV enzymes were assessed to confirm prior findings. We assessed transcriptional changes in hypothalamus (HYP) and hippocampus (HIP) for markers of OS, microglial activity and glucocorticoid signaling using qPCR. Although there were no O_3_ or sex-related differences in the cerebellar activities of OS and mitochondrial enzymes, the levels of protein carbonyls and complex II activities were higher in females regardless of O_3_. There were no statistical differences in baseline expression of genes related to OS (*Cat*, *Dhcr24*, *Foxm1*, *Gpx1*, *Gss*, *Nfe2l2*, *Sod1*) except for lower HYP *Sod1* expression in air-exposed females than males, and higher HIP *Gss* expression in O_3_-exposed females relative to matched males. Microglial marker *Aif1* expression was higher in O_3_-exposed females relative to males; O_3_ inhibited *Itgam* only in males. The expression of *Bdnf* in HIP and HYP was inhibited by O_3_ in both sexes. Genes related to glucocorticoid signaling (*Fkbp4*, *Fkbp5*, *Hsp90aa1*, *Hspa4*, *nr3c1*, *nr3c2*) showed sex-specific effects due to O_3_ exposure. Baseline expression of HIP *Fkbp4* was higher in females relative to males. O_3_ inhibited Nr3c1 in female HIP and male HYP, but *Nr3c2* was inhibited in male HYP. *Fkbp4* expression was higher in O_3_-exposed females when compared to matched males, whereas *Fkbp5* was expressed at higher levels in both brain regions of males and females. These results indicate that sex-specific brain region responses to O_3_ might, in part, be caused by OS and regulation of glucocorticoid signaling.

## 1. Introduction

Sex is a biological variable with significant influence over various physiological functions within all species. Specifically, with regard to lung physiology and immune responses, the factor of sex greatly affects the prognosis of pulmonary morbidities in response to air pollution [1,2,3,4]. Historically, within the literature, causal relationships have been made between air pollution and morbidity and/or mortality by various factors such as genetic susceptibility, environment, and even socioeconomic status. However, little attention has been given to sex differences, especially in animal models. A recent study in Canada was able to analyze nearly three decades of mortality data and hospitalizations along with health outcomes from a geographically diverse cohort representing nearly half of the country as they relate to health effects resulting from air pollution exposure [4]. Shin and colleagues found that, in response to short term exposure to ambient air pollution, there were pollutant season-specific sex differences in circulatory and respiratory hospitalization and mortality where males were at higher risk for warm season, but females were at higher risk than males for cold season [4,5,6]. Given global trends in abnormal climate fluctuations, there is growing concern of human exposure to air pollutants, namely ozone (O_3_) which is produced at ground level by excessive heat and radiation reacting with volatile organic compounds and nitric oxide (NO_X_) gases. A recent multinational report has identified pollution as the leading environmental cause of premature death and disease [7]. Of all types of environmental pollutants, this report identified air pollution to be the primary driver of this trend. Therefore, understanding the effect of gender on adverse health outcomes related to air pollution exposure is paramount.

The immunological and physiological consequences of inhaled toxicants, such as air pollution, have been extensively investigated [5,8,9,10,11,12]. The outcomes include changes in body composition, core temperature, cardio-pulmonary effects such as heart rate, pulmonary sensitivity, and neuronal effects such as motor activity deficits. However, those lines of investigation are now converging on a significant role for the aptly named “lung-brain-axis” and its role in extrapulmonary morbidities of air pollution exposure [13,14]. It has become clear that via a yet unknown mechanism (referred to as the lung-brain-axis), an inhaled toxicant such as O_3_ can generate a physiological stress response through the activation of the hypothalamic-pituitary-adrenal (HPA) axis similar to physical and psychosocial stressors [15,16,17,18,19,20,21,22,23,24]. Sex-dependent differences in HPA axis processing and responses have been previously established [25,26,27]. In this study, we sought to investigate sex-dependent differences in transcriptomic responses in brain regions, specifically the hypothalamus (HYP) and hippocampus (HIP), of animals exposed to O_3_ to understand the effects on HPA axis. We first present data on oxidative stress (OS) status and mitochondrial complex enzymes across sex in the cerebellum in order to demonstrate consistency with previous reports [5,28,29,30,31,32]. These data are then accompanied by novel gene expression data in two important brain regions for the HPA axis (i.e., HYP and HIP). In these brain regions, we assayed genes related to glucocorticoid signaling, microglia, general stress, and oxidative stress in male and female rats acutely exposed to O_3_.

## 2. Results

The effects of O_3_ exposure on the brain have garnered interest in recent years [20,33,34,35,36,37]. O_3_ itself is a highly reactive oxidizer, which does not translocate distally in its parent chemical form. However, many of the effects in the brain seem to be mediated by indirect modes of action, such as by sensory reception or other signaling cascades [13,38]. Due to the chemical nature of O_3_, OS as an adverse outcome is most typically associated with exposure [33,39,40,41].

### 2.1. Effects on OS Parameters in the Cerebellum

We have previously reported O_3_ effects on OS and mitochondrial complex enzymes in different brain regions of various rat species, different age groups, and in combination with other non-chemical stressors [33,34,36,37]. In association with O_3_-induced brain genomic effects in this species and at this age, we have conducted studies in these two parameters in the cerebellum to confirm previous observations (Figure 1 and Figure 2). Although the focus of this study is the sex differences within the integrating and regulatory brain regions (HYP and HIP) through genomic changes, we assayed OS parameters and complex enzyme activities in this proxy region of interest because of tissue limitation. Firstly, NAD(P)H: Quinone dehydrogenase 1 (NQO1; Figure 1A) and NADH ubiquinone reductase (UBIQ; Figure 1B), which were used as indicators of ROS production, were upregulated in O_3_-exposed males when compared to air groups. However, these elevations were minor (43% increase), and a two-way ANOVA failed to indicate a significant difference from air-exposed male animals compared to female animals. For both ROS parameters, the effects were nearly identical in females when exposed to air or O_3_. Superoxide dismutase (SOD), which is an endogenous antioxidant (Figure 1C), showed similar effects in males and females, indicating no observable differences across sex or exposure. Glutathione is another endogenous antioxidant, and γ-glutamylcysteine synthase (γ-GCS) is a key rate-limiting enzyme necessary to produce glutathione. Similar to the activity patterns of SOD, there were no discernable differences in γ-GSC activity across sex and exposure groups (Figure 1D). Total antioxidant substances (TAS) were measured to ensure a complete picture of the antioxidant capacity of the brain region. Just as with the other ROS measures, there were no differences across sex or exposure groups (Figure 1E). OS can have numerous deleterious effects to surrounding cellular structures [42]. These effects are primarily caused by the oxidation of important biomolecules, such as proteins. Protein carbonyl formation is a common oxidation product due to OS and this results in a loss of protein function [43]. Therefore, we measured protein carbonyls as an indicator of oxidative damage produced by the imbalance between ROS production and cellular antioxidant homeostasis. Despite the lack of differences in ROS measures and antioxidant capacities, there were significant differences attributed to sex in the formation of protein carbonyls (F_1,21_ = 6.143; *p* < 0.05) where, compared to their corresponding male exposure groups, females had higher levels of protein carbonyl content (Figure 1F). These results indicate that antioxidant systems that typically respond to ROS production were not upregulated with O_3_ exposure, but that the oxidized biomolecule landscape is different across sex.

### 2.2. Effects on Complex Enzyme Activities in the Cerebellum

Mitochondria are the main sources of endogenous OS since they consume oxygen for energy metabolism. Therefore, the key components of mitochondrial energy production, complex enzymes, were assayed. We assayed three of the five complex enzymes (Figure 2) that comprise the electron transport chain (ETC): NADH: Ubiquinone oxidoreductase (Complex Enzyme I), succinate dehydrogenase (Complex Enzyme II), and cytochrome c oxidase (Complex Enzyme IV). Complex I enzyme activity was similar across both sex and exposure groups (Figure 2A). Females had higher activity levels of complex II compared to males (Figure 2B; F_1,21_ = 24.4; *p* < 0.001). Both complex I and II are entry points to the ETC and these sex differences could indicate sexually dimorphic energy metabolism strategies. The final energy harvesting enzyme in the ETC is complex IV (Figure 2C). There appeared to be an O_3_-mediated increase in females, but two-factor ANOVA did not indicate a statistical difference. Taken all together, these results indicate that there may be differences in mitochondrial energy metabolism strategies between males and females, but minimal influence of acute O_3_ exposure.

### 2.3. Effects on Gene Expression in HYP and HIP

The focus of O_3_ exposure studies has started to shift from analysis of OS and immunological response towards the central control of the responses [20]. Therefore, a panel of neuroendocrine stress, OS-related, microglial, and glucocorticoid gene markers were assayed (Figure 3, Figure 4, Figure 5 and Figure 6). Significantly modified genes showing O_3_ effects and sex-dependent differences is shown in Table 1. A two-way ANOVA revealed no significant differences between sex or O_3_ exposure within the HIP and HYP (Figure 3A). The expression of genes which encodes the receptor for the product of *Adcyap1*, *Adcyap1r1*, was higher in HIP of air-exposed males when compared to air-exposed females. However, no changes were noted due to O_3_ exposure in either HIP or HYP (Figure 3B). No significant exposure, brain region or sex-related differences were noted in *Crhr1* expression (Figure 3C). *Bdnf* expression levels were lower in the HIP of male and female rats (Figure 3D; F_1,18_ = 15.43; *p* < 0.05). In the HYP, there was also a decrease in *Bdnf* expression, however, post hoc multiple comparisons only revealed a significant difference in the females.

Genes related to oxidoreductive processes were also investigated. Although there were no significant exposure- or sex-related differences in the expression of *Gpx1* in either brain region, there was a significant interaction of sex and exposure on *Gss* expression where expression seemingly decreased in males with O_3_ exposure but increased with O_3_ exposure in females (Figure 4B; F_1,14_ = 6.235; *p* < 0.05). In the HYP, there was a significant effect of exposure (F_1,14_ = 6.324; *p* < 0.05), where males had higher expression of *Gss* in response to O_3_ exposure with no change in females. There were no changes in HIP or HYP *Nfe2L2* expression in males or females with or without O_3_ exposure (Figure 4C). There was significant sex-related difference in expression of *Sod1* in the HYP where air-exposed males had higher expression when compared to females (Figure 4D; F_1,17_ = 5.793; *p* < 0.05). In the HIP, *Cat* gene expression tended to be higher in O_3_-exposed males and females, but these differences did not reach significance (Figure 4E). Lastly, the expression levels of *Foxm1* were not affected by sex or O_3_ exposure in either brain region (Figure 4E).

We have previously reported changes in reactive microglia as a result of O_3_ exposure [34,36]. In those studies, we used immunohistochemistry targeting Iba1 (encoded by the gene *Aif1*), to identify microglia in brain sections and then classified them as reactive based on their morphology. In the current study, we assessed gene expression related to microglia function to evaluate if they were transcriptionally regulated and/or showed any sex differences in microglia (Figure 5). Females in general had higher expression of *Aif1* in the HYP regardless of air or O_3_ exposure (Figure 5A; F_1,18_ = 10.16; *p* < 0.05). HYP but not HIP *Itgam* expression was lower in males exposed to O_3_ when compared to the air group. No sex-related differences were noted either in HIP or HYP (Figure 5B). Two-way ANOVA revealed no significant interaction of sex and exposure on *P2ry12* expression (Figure 5C). *Tmem119* is a recently discovered gene specific to microglia [44]. *Tmem119* expression was neither significantly different between sexes nor exposure in either of the brain regions (Figure 5D).

Acute O_3_ exposure has been shown to increase glucocorticoids systemically [35], and these levels were linked to O_3_ adaptation [45]; therefore, we also analyzed genes related to glucocorticoid signaling. The gene that encodes for the glucocorticoid receptor, *Nr3c1*, showed a significant interaction of both sex and exposure in each brain region (F_1,17_ = 7.431; *p* < 0.05) where only females had lower expression in response to O_3_ exposure in HIP; whereas, in HYP, O_3_-exposed males had lower expression when compared to air controls (Figure 6A). The gene for the mineralocorticoid receptor, *Nr3c2*, showed no changes by either sex or O_3_ exposure in HIP, but in HYP males exposed to O_3_ had lower expression relative to air controls (Figure 6B). One of the chaperone proteins that is a part of the glucocorticoid receptor heterocomplex is heat shock protein 90, encoded by *Hsp90aa1* (Figure 6C). In both brain regions, although there were no apparent effects of O_3_ exposure on *Hsp90aa1* expression, the female rats exhibited lower expression when compared to males in HYP, but these differences did not reach significance. No sex- or exposure-related differences were noted in expression of *Hspa4* in either brain region (Figure 6D). Two other members of the glucocorticoid receptor heterocomplex that play roles in the translocation of the complex upon binding of stress hormones are the gene products of *Fkbp4* and *Fkbp5*. In HIP, there was a significant effect of sex (Figure 6E; F_1,16_ = 7.933; *p* < 0.05) resulting in overall higher expression levels in females exposed to air compared to males. In the HYP, there was a significant interaction of sex and exposure where females had higher overall expression of *Fkbp4*, and expression further increased in females with exposure to O_3_. There were significant effects of O_3_ exposure in females and in both brain regions on *Fkbp5* expression (Figure 6F; HIP: F_1,17_ = 9.247; *p* < 0.05; HYP: F_1,17_ = 16.63; *p* < 0.05). *Fkbp5* expression was also higher in O_3_-exposed males when compared to air-exposed males, but these differences did not reach significance.

## 3. Discussion

O_3_ is a ubiquitous air pollutant and has been extensively studied for its neurological and neuroendocrine impacts from acute and long-term exposures [46,47]. O_3_ is highly reactive and not likely to translocate to distant organs, such as the brain, upon inhalation; however, numerous studies have shown that O_3_ exposure induces OS in multiple brain regions [33,34,48], causes transcriptional changes [35], increases inflammatory phenotype [49] and microglial activation [50], and in the long-term promotes Alzheimer’s disease-like pathology in laboratory rodents [47]. More recently exposure to O_3_ has been shown to activate neuroendocrine stress pathways such as the HPA, leading to increased levels of glucocorticoids in circulation [19,51], however, sex differences and neurological effects are not well-examined in relation to neuroendocrine modulation. In this study, we hypothesized that O_3_-induced transcriptional changes in OS markers will be associated with markers involved in glucocorticoid signaling and these changes will be sex-dependent.

This study demonstrates that cerebellar protein carbonyl levels and mitochondrial complex II activities were higher in females regardless of O_3_, but the activities of antioxidants were unchanged between sex or exposure condition. Overall gene expression changes in HIP and HYP of male and female rats indicated inhibition of *Bdnf* by O_3_ in HIP of both sexes and only HYP in males. O_3_ increased *Gss* expression in female HIP and male HYP, suggesting the potential contribution of glutathione in modulating brain region-specific effects. Microglial marker *Aif1* had high baseline expression in HYP of females when compared to males and in the same region the other marker *Itgm* was inhibited by O_3_ but only in males. These sex-specific changes in OS, markers were associated with changes in markers of glucocorticoid signaling. We noted that glucocorticoid receptor, *Nr3c1* was transcriptionally inhibited by O_3_ in HIP in females, and in HYP but only in males. The expression of mineralocorticoid receptor, *Nr3c2* was also inhibited by O_3_ (HYP > HIP) but only in males. The expressions of glucocorticoid chaperone proteins were also affected by O_3_ in sex-specific manner. Baseline expression of *Fkbp4* was higher in females and chaperone protein encoded by *Fkbp5*, critical in regulating glucocorticoid activity, was induced by O_3_ in both brain regions (females > males), suggesting sex-dependent temporal regulation of HIP and HYP glucocorticoid activity by O_3_. Collectively, these data show that glucocorticoids known to be increased during O_3_ exposure [51] might regulate brain region-specific effects on stress dynamic and oxidation reduction processes in sex-dependent manner.

We have recently demonstrated that neural, pulmonary, and systemic effects of O_3_ were mediated through the activation of sympathetic adrenal medulla (SAM) and HPA, associated with reduction of pituitary hormones regulating gonadal axis in males [35,45,51], and that diminution of circulating adrenal hormones through adrenalectomy resulted in the lack of O_3_-induced reduction of gonadal hormones. This suggests that adrenal hormones could centrally regulate O_3_ effects on pituitary gonadal hormones [35]. This also implies that O_3_ neuroendocrine effects are likely sex-dependent, however, sex differences in O_3_ neuroendocrine effects are not well examined. Here we wanted to focus on glucocorticoid-mediated neural changes in male and female rats, since this adrenal hormone has been shown to regulate longevity and characteristics of a stress phenotype through brain region and sex-dependent differential regulation of cellular activities [52,53]. Alterations in glucocorticoid mechanisms and their levels are associated with chronic stress-related neurological disorders, known to affect males and females differently [54].

Impacts on oxidoreductive processes such as levels of oxidation byproducts and changes in oxidoreductive enzyme activities after O_3_ exposure have been demonstrated in various brain regions of rodents [33,34,48]. Although it is not entirely clear if O_3_ effects are a general oxidative response across all regions, or specific to a given brain region, the available data supports brain region-specific changes [33,36]. Some of these studies have examined sex differences in O_3_ effects [55]. Here, we wanted to use cerebellar tissues to determine concordance with our prior studies reporting oxidative and mitochondrial bioenergetic changes after O_3_ exposure in rats [33,34,36] and evaluate sex differences in parallel in the same study. Protein carbonyl contents in 4-month but not 12-month old male Brown Norway rats exposed to 1 ppm O_3_ for two days was increased in our prior study [33]. Phenotypic oxidative changes were also noted in other brain regions of rats after O_3_ exposure [33,36], but in this study only the expression of markers linked to OS, mitochondrial activity, and microglia were examined in other brain regions, specifically HIP and HYP. Acute O_3_ exposure was associated with increased lipid peroxidation in male HIP and cerebellum of rats exposed to O_3_ [56]. Thus, O_3_ exposure is associated with oxidative changes in various brain regions of rats, and our data show that females might have greater susceptibility for cerebellar oxidative changes when compared to males.

Transcriptionally regulated genes involved in responding to underlying OS might be differentially expressed between males and females in two stress responsive regions of the brain, likely mediating neural and systemic response to SAM and SAM axes activation by O_3_ [35,51]. *Gss* encodes for a critical intracellular antioxidant, glutathione, critical in protecting the brain from oxidative processes [57], and was induced after O_3_-exposure in female HIP and in male HYP. This suggests sex- and brain region-specific roles in cellular oxidoreductive mechanisms involving transcriptional upregulation, likely in response to increased oxidative cell stress. Interestingly, this was associated with higher baseline *Sod1* transcription in HYP of males when compared to females, which suggests that HYP in males might be more susceptible to some stressor-induced changes when compared to females.

The expression of *Bdnf* involved in neuropsychological disease [58] was inhibited by O_3_ in HIP of both sexes and in HYP of only males. *Bdnf* has been suggested to play a role in not only regulating stress mechanisms but also in glucocorticoid signaling [59]. While glucocorticoids are increased in the circulation after O_3_ exposure, the interactive roles of *Bdnf* and glucocorticoids have been reported in humans [60]. Lower BDNF protein levels were noted in HIP of rats with compressed dorsal root ganglion and exposure to O_3_ [61]. O_3_-induced reduction in circulating BDNF levels was mitigated in rats treated with metyrapone, a glucocorticoid synthesis inhibitor, suggesting a regulatory role of glucocorticoid in mediating transcriptional repression in HIP of male rats exposed to O_3_ [62]. Interestingly, the effect in HYP of females was smaller compared to males in our study. The overall impact on *Bdnf* modulation by O_3_ suggests its involvement in long-term regulation of stress dynamicity.

Microglia in different brain regions play a critical role in maintaining homeostatic processes by responding to stress and inducing appropriate immunological processes [63]. Microglial activation has been reported by a variety of stressors including O_3_ [14,64,65]. We wanted to assess sex-related differences in microglial activity and potential acute O_3_-mediated stress effects. Although there were no O_3_ effects on transcriptional activation of microglial marker *Aif1* (also known as *Iba1* at transcriptional level), the increased AIF1 immunoreactivity has been noted after O_3_ exposure in rodents [14]. However, we noted that the baseline levels of *Aif1* transcription were higher in female HYP when compared to males. Moreover, we noted that in male but not female HYP, O_3_ exposure lowered mRNA expression of *Itgam* (complement receptor 3). *Itagm* has been shown to be expressed on microglia and plays a critical role in neurodegenerative processes [66]. The significance of transcriptional repression of this gene after O_3_ exposure is unclear but in chronic exposures might contribute to long-term neurological disorders associated with neuroinflammation.

One of the ancillary objectives of this study was to compare O_3_ response on markers of glucocorticoid signaling in HIP and HYP of male and female rats based on our prior demonstration of involvement of dynamic glucocorticoids in O_3_-mediated neural effects, release of pituitary hormones and especially adaptation response noted after repeated O_3_ exposure [35,45,51]. Brain region- and sex-specific inhibition of glucocorticoid receptor gene, *Nr3c1*, activated by high levels of circulating glucocorticoids, showing O_3_-induced inhibition in HIP of females and HYP of males. This coincides with inhibition of *Nr3c2*, a mineralocorticoid receptor activated by low levels of glucocorticoids in HYP of males, might reflect complex glucocorticoid regulating mechanisms. The differences in receptor density and the glucocorticoid activity regulating mechanisms between these brain regions [67,68] and sex in response to stressors [69] might underlie observed differences in O_3_-induced transcriptional changes in glucocorticoid receptors [57,70,71]. *Nr3c1*, a glucocorticoid receptor gene and *Nr3c2*, a mineralocorticoid receptor gene, are activated by circulating glucocorticoids entering cells in the brain centers and are involved in finetuning the dynamic of a stress response in association with glucocorticoid scaffolding protein encoded by *Fkbp5*. This fine-tuning of glucocorticoid transcriptional action can be accomplished by relative cell and brain region-specific distribution and transcriptional activity of these receptors and scaffolding proteins, since nearly 20% of the human genome is regulated by these glucocorticoids critical in responding to environmental stressors through changes in physiological processes [70]. The activity of glucocorticoids is also regulated by androgen sex hormones, such as estrogen through its interactive transcriptional regulation [71]. Thus, while the precise mechanisms of how sex hormones might regulate glucocorticoid effects in ozone-induced glucocorticoid receptor transcriptional changes are likely complex and cannot be ascertained through this observational study, here we show that ozone exposure might mediate sex specific changes.

Two FK506-binding immunophilins, encoded by *Fkbp4* and *Fkbp5* genes comprise an important part of glucocorticoid receptor heterocomplex that plays a role in the regulation of transcriptional activity of glucocorticoids [72,73]. *Fkbp5* bound to glucocorticoid receptor is released from the complex upon binding to glucocorticoids, allowing *Fkbp4* to bind to the hormone-bound transcriptional complex mediating transcriptional activity through glucocorticoid response elements (GRE) of hundreds of genes including *Fkbp5.* In turn, *Fkbp5* rapidly binds to the cytosolic receptor complex and regulates glucocorticoid activity through a feedback loop [74]. We have shown that O_3_ exposure leads to HPA-mediated glucocorticoid release in the circulation within minutes of exposure [35,45,51]. We noted that there were sex- and brain region-specific effects of O_3_, likely mediated by higher circulating glucocorticoids on expression of *Fkbp4* and *Fkbp5*. We noted that females have a high baseline level of mRNA expression of *Fkbp4* but not *Fkbp5* in both brain regions. The precise role of estrogen in regulating glucocorticoid activity cannot be ascertained, but it has been shown that chaperone proteins encoded by *Fkbp4* and *Fkbp5* genes can regulate the stability of estrogen receptor alpha [75] and might be influencing the transcriptional activity of these proteins. Transcriptionally regulated *Fkbp5* was induced by O_3_ in both sexes and brain regions, albeit at different levels, suggesting that circulating glucocorticoids modulate O_3_ stress effects in the brain and might be involved in adaption response through glucocorticoid signaling [45]. *Fkbp5* polymorphisms have been associated with a variety of human psychiatric ailments [73]. These data provide the link between air pollution and neural stress-mediated glucocorticoid changes and highlight the importance of neuroendocrine mechanisms in chronic neural diseases that have been recently linked to high air pollution levels [76].

There are a number of limitations in this study. This observational study comparing sex differences in O_3_-induced OS and glucocorticoid-mediated transcriptional changes neither assessed potential protein levels of markers or functional effects, and rather associated changes to our prior publications related to neural and glucocorticoid changes. The O_3_ concentrations used are several folds higher than what is encountered environmentally in tropical atmospheres, however, they are comparable to human clinical studies performed during intermittent exercise with O_3_ concentrations ranging from 0.2 to 0.4 ppm [77,78,79,80]. Humans exposed during intermittent exercise retain 4–5 times the inhaled dose of O_3_ compared to rodents exposed during rest [81]. These data highlight the sex differences and regulatory roles of neuroendocrine pathways in mediating neural effects of O_3_, while much of mechanistic sex differences will need to be further examined.

## 4. Materials and Methods

### 4.1. Animals

Long–Evans (LE) rat dams were obtained from Charles River Laboratories, Inc. (Raleigh, NC, USA) and maintained in the animal colony. After birth, the litters were standardized on postnatal day (PND) six to four males and four females where possible. The sex was determined by measuring anogenital distance, where it is more for males than females. If four males and four females were not possible, then any combination of eight pups per litter was allowed. At PND 21, they were weaned two males or two females per cage and raised in the animal facility until O_3_ exposure. The animal rooms are maintained at 21 °C, 55–65%, relative humidity and 12 h light/dark cycle. The EPA animal facility is approved by the Association for Assessment and Accreditation of Laboratory Animal Care. Throughout the acclimation and experimentation period, animals were fed Harlan TD.08806 rat chow (Harlan Teklad Global Diet) and drank tap water, *ad libitum*. U.S. EPA’s Animal Care and Use Committee approved the experimental protocol and the guidelines of the National Institutes of Health (NIH) for the care and use of rats were followed (NIH Publications No. 8023).

### 4.2. Ozone Generation & Exposure

When the rats reached an age of ~160 days, they were randomly assigned to a two-day exposure to filtered air or O_3_ (0.8 ppm for 4 h/day for two consecutive days). Both male and female rats were examined for ozone effects using filtered air or O_3_ exposures. O_3_ was generated from oxygen by a silent arc discharge generator (OREC, Phoenix, AZ, USA), and its entry into the Rochester style “Hinners” chambers was controlled by mass flow controllers. The O_3_ concentrations in the chambers were recorded continuously by photometric O_3_ analyzers (API Model 400, Teledyne Instruments; San Diego, CA, USA). Air temperature and relative humidity were monitored continuously. The measured levels of O_3_ in each chamber were within ±0.01 ppm of the targeted concentration. O_3_ had no effect on environmental variables; air temperature and relative humidity in the chambers were maintained at 23–24 °C and 50%, respectively. The rats were placed in individual stainless-steel wire-mesh exposure cages (27.3 cm long × 14.6 cm wide × 7.75 cm tall) which were part of a 16-cage unit. Animals were exposed to HEPA-filtered room air (0 ppm) or 0.8 ppm O_3_ for 4 h/day. Exposure began at ~0615 h each day and necropsies were performed after completion of second day exposure. The selected O_3_ concentration of 0.8 ppm is several folds higher than what has been achieved in the areas of the United States [79]. However, 0.8 ppm O_3_ exposure in resting rats is comparable to clinical studies using 0.2–0.3 ppm O_3_ exposure in humans during intermittent exercise based on the airway dose deposition [81]. At these concentrations, O_3_ is known to induce cardiovascular effects in humans and mild airway inflammation in humans and animals [18]. In metropolitan and tropical areas, O_3_ concentrations of 0.1 to 0.4 ppm have been reported [77,78]. Using 0.8 ppm concentration in a small subset of rats allowed us to characterize acute effects of O_3_ in the brain regions and compare sex differences.

### 4.3. Necropsy & Tissue Isolation

Rats were euthanized within 1–2 h after the second O_3_ exposure with an intraperitoneal injection of >200 mg/kg sodium pentobarbital (Fatal-Plus diluted 1:1 with saline; Vortech Pharmaceuticals, Ltd., Dearborn, MI, USA). When animals were completely nonresponsive to hind paw pinch after Fetal-Plus injection, abdominal aorta blood samples were collected. Blood was collected directly into vacutainer tubes containing EDTA for complete blood counts and without EDTA for serum preparation. The results from blood samples were published separately [31,32]. Brains were quickly removed, and brain regions (cerebellum, hippocampus, and hypothalamus) were dissected on ice [82], quick frozen on dry ice, and stored at −80 °C until analyzed.

### 4.4. Tissue Preparation for OS Measures in Cerebellum

The brain cerebellum samples were homogenized with a polytron in 20 mM cold Tris-HCl buffer (pH 7.4) at 50 mg/mL and centrifuged at 8000× *g* for 20 min at 4 °C. To evaluate ROS production, the activities of NAD(P)H: Quinone oxidoreductase (NQO1) and NADH-Ubiquinone reductase (UBIQ-RD) were measured. NQO1 activity was calculated from the difference in reaction rates of the NADH and menadione-dependent dicumarol-inhibitable reduction of cytochrome C obtained with and without dicumarol. An extinction coefficient of 18.5 mM^−1^cm^−1^ was used in calculations of specific activity [83]. The UBIQ-RD activity was assayed by monitoring the oxidation of NADH+, with the ultimate reduction of ubiquinone to ubiquinol [84]. The rate of UBIQ-RD activity was measured as rotenone-sensitive rate of NADH oxidation at 37 °C and 340 nm. Total antioxidant status was measured using a kit from RANDOX Laboratories (Crumlin, Co., Antrim, UK). ABTS^®^ (2,2′-Azino-di-[3-ethylbenzthiazoline sulphonate]) and was incubated with a peroxidase (metmyoglobin) and H_2_O_2_ to produce the free radical cation ABTS^®^+ which has a stable blue-green color, (600 nm). Superoxide dismutase (SOD) activity was measured using a kit from RANSOD (Randox Laboratories, Oceanside, CA, USA). SOD catalyzes the reaction of superoxide radicals to oxygen and hydrogen peroxide. In this assay, xanthine and xanthine oxidase are used to form superoxide radicals, which react with 2-(4iodophenyl)-3-(4-nitrophenol)-5-phenyltetrazolium (INT) to form a red dye. SOD activity measured spectrophotometrically was determined by the degree to which this reaction was inhibited. γ-Glutamyl cysteine synthetase activity was determined from the rate of formation of ADP (assumed to be equal to the rate of oxidation of NADH) as calculated from the change in absorbance at 340 nm. Coomassie Plus Protein Assay Kit (Pierce, Rockford, IL, USA) and BSA standards from Sigma Chemical Co. (St. Louis, MO, USA) were used to determine protein concentration in the cerebellar tissue extract. All these assays were modified and adapted for use on the KONLAB clinical chemistry analyzer (Thermo Clinical Lab Systems, Espoo, Finland).

### 4.5. Markers of Cellular Damage

Protein carbonyls were assayed using commercial kits from Cayman Chemical Company (Ann Arbor, MI, USA). This assay kit utilizes the 2,4,-dinitrophenylhydrazine (DNPH) reaction to measure the protein carbonyl content in a convenient 96-well format. The amount of protein-hydrazone produced was quantified spectrophotometrically at an absorbance between 360–385 nm.

### 4.6. Assessment of Cerebellum Mitochondrial Complex I, II, and IV Enzyme Activities

ELISA kits for complex enzymes I, II, and IV (Abcam, Cambridge, MA, USA: #AB109721, #AB109908, and #AB109911, respectively) were used to determine enzyme activities in each brain region. Briefly, Complex I activity was quantified by measuring the oxidation of NADH to NAD+ and simultaneous reduction of dye, which increases absorbance at 450 nm. The mitochondrial Complex II assay kit catalyzes electron transfer from succinate to the electron carrier ubiquinone. The production of ubiquinone is coupled to reduction of dichlorophenolindophenol dye, causing it to become colorless and decrease in absorbance at 600 nm. Mitochondrial Complex IV is quantified by measuring the oxidation of reduced cytochrome c, which yields a decrease in absorbance at 550 nm. Absorbance was determined on 96-well plates read on a SpectraMax M5 spectrophotometer running SoftMax ProV5 software (Molecular Devices, San Jose, CA, USA). The reaction rates (Vmax) were calculated from the linear portion of the output curve. All values were standardized by expressing as activity/mg protein as determined by BCA (Thermo Scientific, Rockford, IL, USA).

### 4.7. Quantitative Polymerase Chain Reaction

Snap-frozen HIP and HYP were used to isolate RNA via spin column-based isolation kit (Qiagen Rneasy Mini Kit #74104, Germantown, MD, USA). Samples were homogenized with Omni international tissue homogenizer (Omni #59136, Kennesaw, GA, USA) in TriZol (Thermofisher, #15596026, Waltham, MA, USA) then precipitated with chloroform (Sigma, C5312). Seventy percent ethanol was added to the aqueous phase before being processed through the RNeasy spin column. The amount of RNA per sample was quantified using the Qubit 4 Fluorometer (Invitrogen, Q33238, Waltham, MA, USA) and Qubit RNA BR Assay Kit (Invitrogen, Q10211, Waltham, MA, USA) and then diluted to 1 ng/µL with Nuclease free water (Ambion, AM9938, Waltham, MA, USA). The PCR was run using the Bioline SensiFAST™ SYBR^®^ No-ROX One-Step Kit (Bioline, Thomas Scientific, BIO-72005, Waltham, MA, USA). Primers were designed in house and synthesized by IDT (Table 2; Integrated DNA technologies, Coralville, IA, USA). Efficiency curves were performed for all primers and only primer sets with efficiencies of 90–110% were accepted for use in this study. All PCR was performed on the Quant Studio 7 Flex PCR Machine. All PCR reactions underwent melt curve analysis to verify that only single amplicons were amplified. Those that had multiple melt curves, indicating multiple products were synthesized, were removed. For analysis, an R script was written to perform the Pfaffl equation for relative fold gene expression [85]. The Pfaffl equation for differential gene expression takes into consideration the CT values (as in ΔΔCT), and the real-time efficiencies of the primers. Next, we used the Tukey’s Interquartile Range (IQR) outlier test, which uses cutoffs 1.5 IQR as a gating parameter, for each gene within brain regions. Samples that were outliers across more than one gene were removed from the entire dataset, as their values were not reliable. We then used the same Tukey IQR analysis to determine any other potential outliers, which were removed per gene and brain region. It should also be noted that the *Fkbp5* gene demonstrated multiple melt curves for all samples. However, given the robust O_3_ response for *Fkbp5*, we decided to include this gene in the current data set.

### 4.8. Statistical Analysis

Statistical analyses were performed using RStudio [86]. Raw data were organized with the dplyr package [87]. Two-way ANOVA was used to analyze difference in gene expression using the factors of O_3_ exposure and sex. Significant effects and/or interactions were followed with two forms of post-hoc tests looking at main effect differences from O_3_ exposure and sex. To analyze average effects of sex, a Dunnett’s many-to-one multiple comparison test was performed using the trt.vs.ctrl contrast argument of emmeans package [88]. To analyze the effects of O_3_ within each sex, we performed pairwise comparisons using the pairwise contrast argument of emmeans. All figures were generated using the ggplot2 [89] and cowplot [90] packages.

## 5. Conclusions

Supporting the link between exposure to air pollutants and exacerbation of neurological diseases differentially in males and females, here we show that, O_3_, a prototypic oxidant air pollutant, induced sex-specific alterations in phenotypic markers of OS in the cerebellum, together with transcriptional changes in enzymes regulating OS and microglial activation within HIP and HYP. More importantly, we noted that there were major sex-specific changes in mRNA expression of glucocorticoid and mineralocorticoid receptors that were associated with increases in expression of transcriptionally regulated glucocorticoid scaffolding and chaperone proteins, implying that glucocorticoids might regulate sex-dependent neural effects of O_3_ in different brain regions. Given the central role of glucocorticoids in physiological and environmental stress and resiliency, their importance in mediating O_3_ neural effects requires further study.

## Figures and Tables

**Figure 1 ijms-24-06404-f001:**
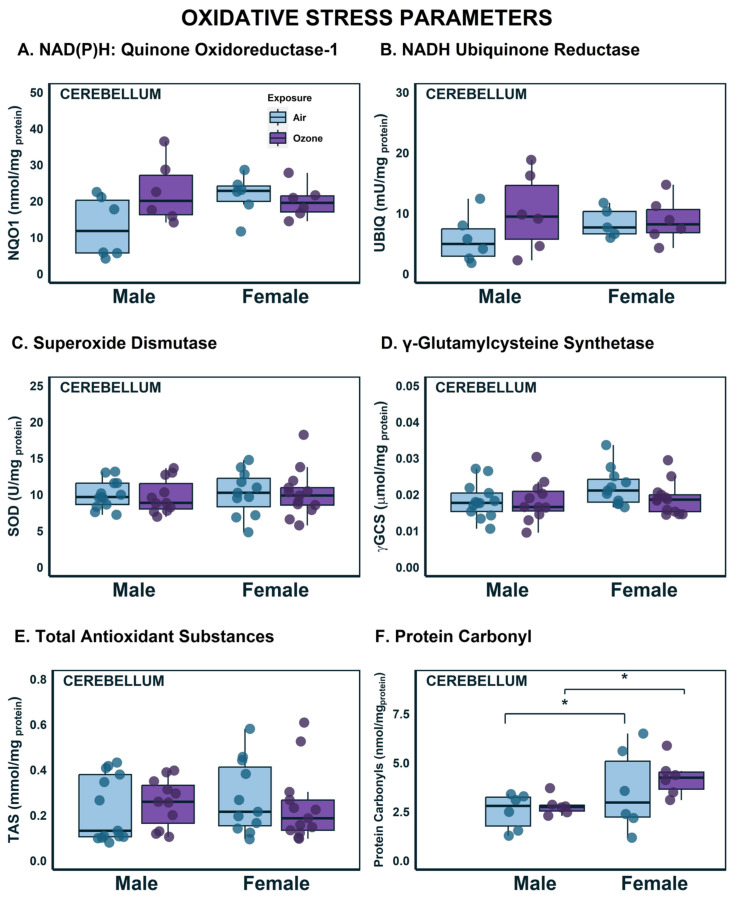
Oxidative stress status in the cerebellum following ozone exposure in male and female rats. (**A**) NAD(P)H Quinone Oxidoreductase-1 activity (n = 6); (**B**) NADH Ubiquinone Reductase activity (n = 6); (**C**) Superoxide dismutase activity (n = 11–12); (**D**) γ-Glutamyl cysteine Synthase activity (n = 11–12); (**E**) Total Antioxidant Substances (n = 11–12); (**F**) Protein Carbonyls (n = 6). * Statistical difference *p* < 0.05. Data are represented by boxplots with a dot plot overlay to express the distribution of each datapoint.

**Figure 2 ijms-24-06404-f002:**
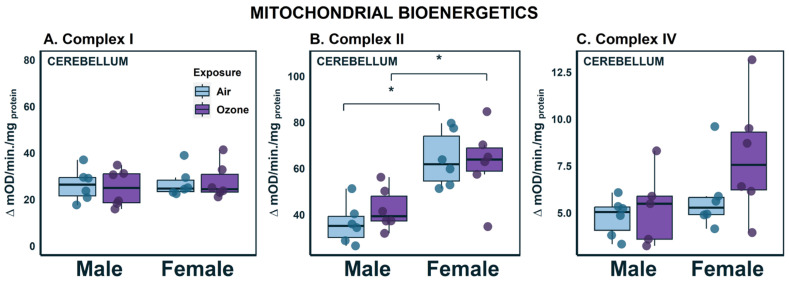
Mitochondrial complex enzyme activities in the cerebellum following ozone exposure in male and female rats. (**A**) Complex Enzyme I activity; (**B**) Complex Enzyme II activity; (**C**) Complex Enzyme IV activity. Data are represented by boxplots with a dot plot overlay to express the distribution of each datapoint where n = 6. * Statistical difference *p* < 0.05.

**Figure 3 ijms-24-06404-f003:**
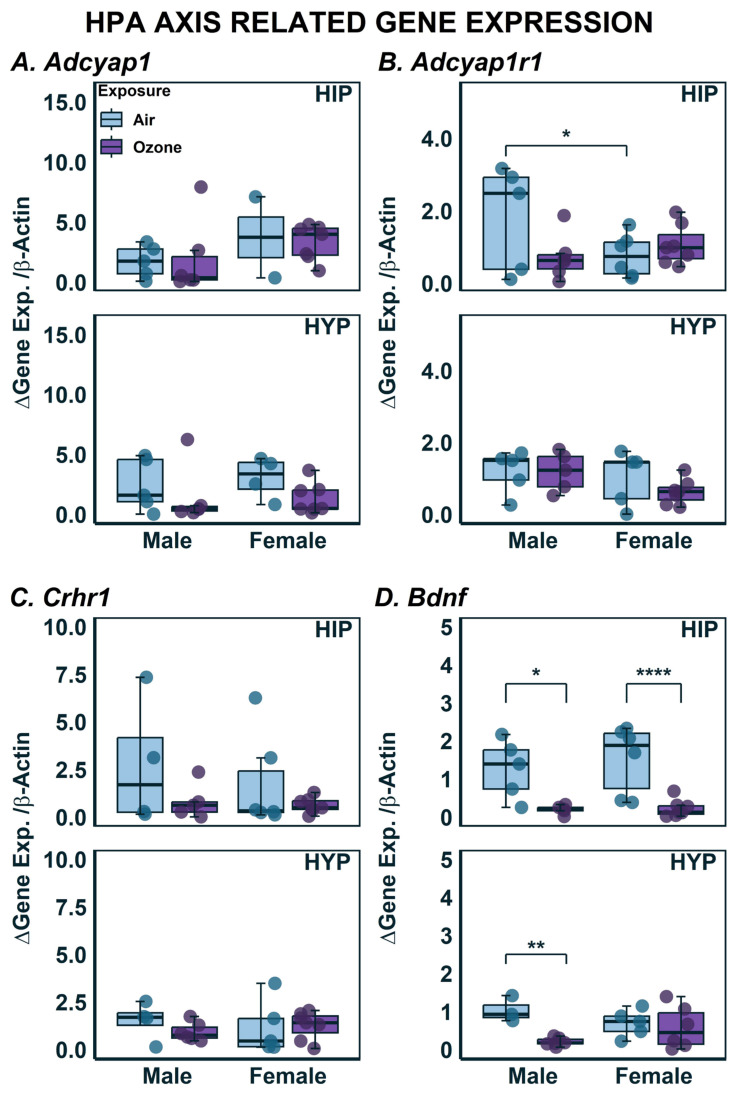
Stress Related Gene Expression in the hippocampus (HIP) and hypothalamus (HYP) following ozone exposure in male and female rats. (**A**) *Adcyap1*; (**B**) *Adcyap1r1*; (**C**) *Crhr1;* (**D**) *Bdnf* expression in HIP and HYP as determined using qPCR. Data are expressed by boxplots where n = 4–6 after removal of outliers using Tukey’s Interquartile Range (IQR) outlier test. * Represents post-hoc statistical differences where *p* < 0.05; ** Represents significance at *p* < 0.01; **** Represents significance at *p* < 0.0001.

**Figure 4 ijms-24-06404-f004:**
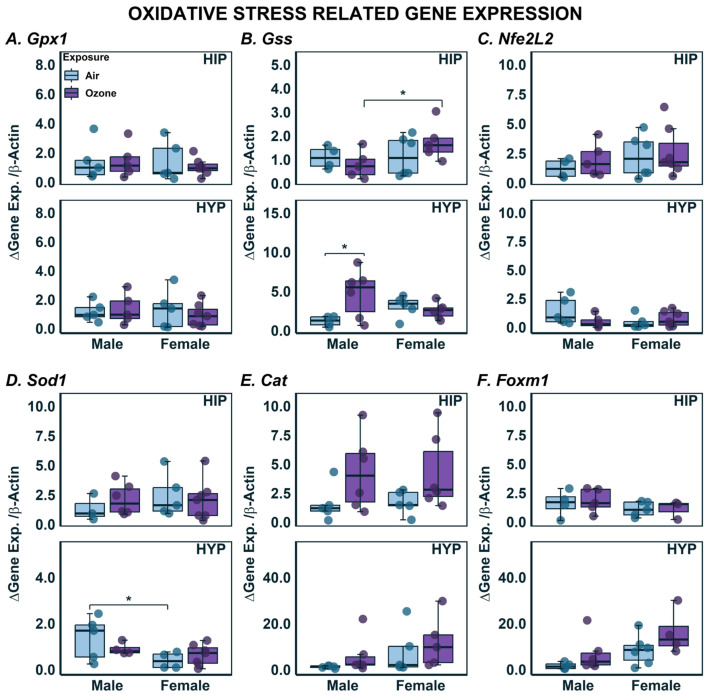
Gene Expression related to oxidative stress in the hippocampus (HIP) and hypothalamus (HYP) following ozone exposure in male and female rats. (**A**) *Gpx1*; (**B**) *Gss*; (**C**) *Nfe2l2*; (**D**) *Sod1*; (**E**) *Cat*; (**F**) *Foxm1* expression was assessed using qPCR in HIP and HYP. Data are expressed by boxplots where n = 4–6 after removal of outliers using Tukey’s Interquartile Range (IQR) outlier test. * Represents post-hoc statistical differences where *p* < 0.05.

**Figure 5 ijms-24-06404-f005:**
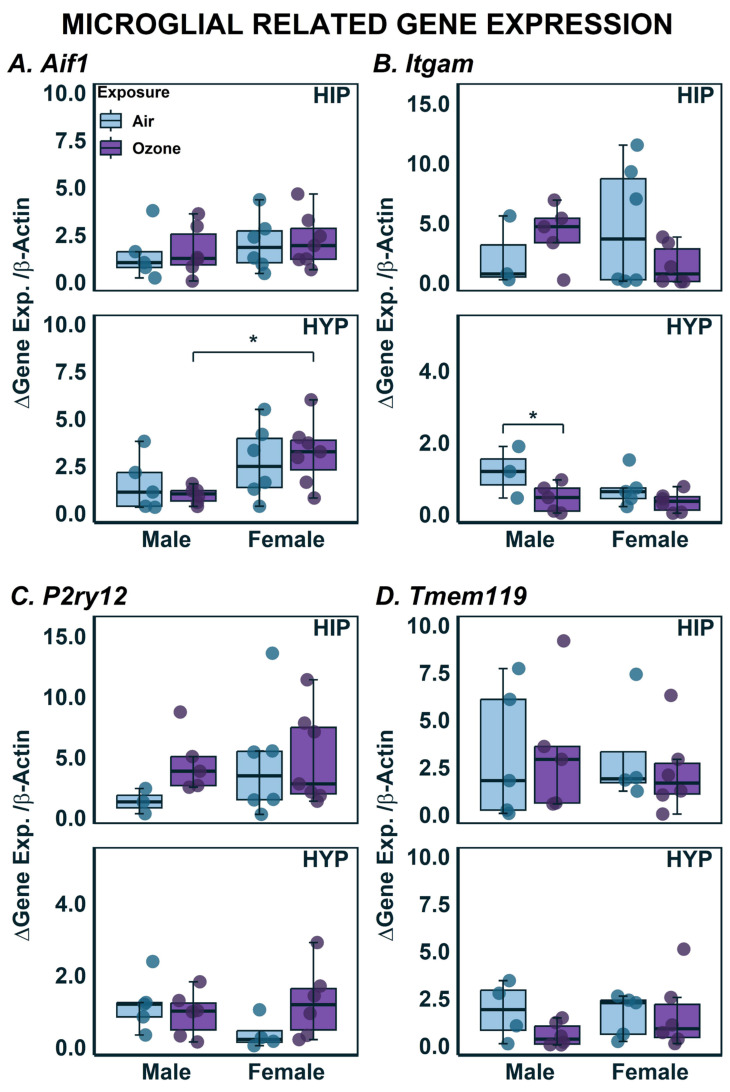
Microglial Associated Gene expression in the hippocampus (HIP) and hypothalamus (HYP) following ozone exposure in male and female rats. (**A**) *Aif1*; (**B**) *Itgam*; (**C**) *P2ry12*; (**D**) *Tmem119* expression was assessed using qPCR in HIP and HYP. Data are expressed by boxplots where n = 4–6 after removal of outliers using Tukey’s Interquartile Range (IQR) outlier test. * Represents post-hoc statistical differences where *p* < 0.05.

**Figure 6 ijms-24-06404-f006:**
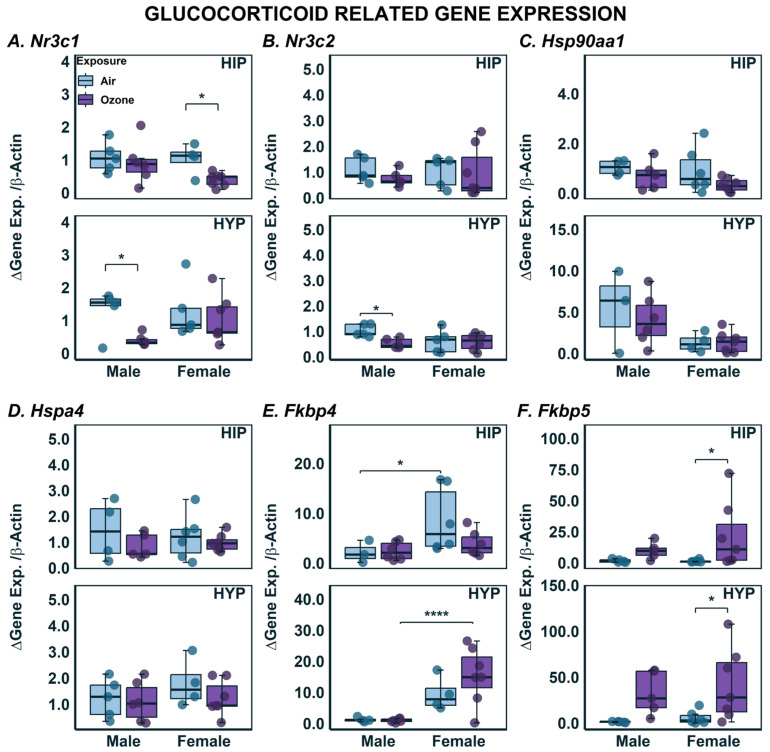
Glucocorticoid Related Gene Expression in the hippocampus (HIP) and hypothalamus (HYP) following ozone exposure in male and female rats. (**A**) *Nr3c1*; (**B**) *Nr3c2*; (**C**) *Hsp90aa1*; (**D**) *Hspa4*; (**E**) *Fkbp4*; (**F**) *Fkbp5* expression was assessed using qPCR in HIP and HYP. Data are expressed by boxplots where n = 4–6 after removal of outliers using Tukey’s Interquartile Range (IQR) outlier test. * Represents post-hoc statistical differences where *p* < 0.05; **** Represents significance at *p* < 0.0001.

**Table 1 ijms-24-06404-t001:** Significantly Modified Genes Showing Ozone Effects and Sex Dependent Changes.

	Hippocampus							
					Male v. Female		Air v. Ozone	
	# Males		# Females		*Significance*		*Significance*	
Gene	*Air*	*Ozone*	*Air*	*Ozone*	*Air*	*Ozone*	*Male*	*Female*
**Glucocorticoid Genes**								
*Fkbp4*	3	6	6	6	3.8 × 10^−2^ (**↑**)	-	-	-
*Fkbp5*	5	5	6	7	-	-	-	2.9 × 10^−2^ (**↑**)
*Nr3c1*	5	6	4	7	-	-	-	4.1 × 10^−2^ (**↓**)
*Nr3c2*	5	5	5	7	-	-	-	-
*Hsp90aa1*	4	5	6	7	-	-	-	-
*Hspa4*	4	5	6	7	-	-	-	-
**HPA Axis Genes**								
*Bdnf*	5	5	6	7	-	-	1.0 × 10^−2^ (**↓**)	9.0 × 10^−4^ (**↓**)
*Adcyap1*	5	6	2	7	-	-	-	-
*Adcyap1r1*	5	6	6	7	-	-	4.3 × 10^−2^ (**↓**)	-
*Crhr1*	4	5	6	7	-	-	-	-
**Microglial Genes**								
*Aif1*	5	6	6	7	-	-	-	-
*Itgam*	3	5	6	6	-	-	-	-
*P2ry12*	3	5	6	7	-	-	-	-
*Tmem119*	5	5	5	6	-	-	-	-
**Oxidative Stress Genes**								
*Cat*	5	6	5	6	-	-	-	-
*Foxm1*	4	5	5	3	-	-	-	-
*Gpx1*	5	5	5	7	-	-	-	-
*Gss*	4	5	6	5	-	4.7 × 10^−2^ (**↑**)	-	-
*Nfe2l2*	4	5	6	7	-	-	-	-
*Sod1*	3	6	5	7	-	-	-	-
	**Hypothalamus**							
**Glucocorticoid Genes**								
*Fkbp4*	4	6	4	7	-	4.0 × 10^−4^ (**↑**)	-	-
*Fkbp5*	5	5	6	7	-	-	-	1.8 × 10^−2^ (**↑**)
*Nr3c1*	5	6	5	7	-	-	2.8 × 10^−2^ (**↓**)	-
*Nr3c2*	5	5	5	6	-	-	2.4 × 10^−2^ (**↓**)	-
*Hsp90aa1*	3	6	4	7	-	-	-	-
*Hspa4*	5	6	4	7	-	-	-	-
**HPA Axis Genes**								
*Bdnf*	3	6	5	6	-	-	7.4 × 10^−3^ (**↓**)	-
*Adcyap1*	5	6	4	7	-	-	-	-
*Adcyap1r1*	5	5	5	7	-	-	-	-
*Crhr1*	4	6	5	7	-	-	-	-
**Microglial Genes**								
*Aif1*	5	6	6	7	-	1.5 × 10^−2^ (**↑**)	-	-
*Itgam*	3	5	5	6	-	-	4.4 × 10^−2^ (**↓**)	-
*P2ry12*	5	6	4	6	-	-	-	-
*Tmem119*	4	6	5	6	-	-	-	-
**Oxidative Stress Genes**								
*Cat*	5	6	5	5	-	-	-	-
*Foxm1*	4	6	6	4	-	-	-	-
*Gpx1*	5	5	5	7	-	-	-	-
*Gss*	4	6	5	6	-	-	1.0 × 10^−2^ (**↑**)	-
*Nfe2l2*	5	5	5	7	-	-	-	-
*Sod1*	5	4	4	7	2.3 × 10^−2^ (**↓**)	-	-	-

Note: **↑** Indicates an upregulation (**red**) and **↓** Indicates a downregulation (**blue**).

**Table 2 ijms-24-06404-t002:** List of PCR Primers.

Gene	Accession Number	Forward Primer Sequence	Reverse Primer Sequence	Product Length	Efficiency	Product Length
*Actb*	NM_031144.3	GTGTGGATTGGTGGCTCTATC	AACGCAGCTCAGTAACAGTC	137	96.184	137
*Adcyap1*	NM_016989.2	GAAGAAGAGGCTTACGATCAGG	GTCCAAGACTTTGCGGTAGG	177	97.664	177
*Adcyap1r1*	NM_001270579.1	GGAAGTGAGGTCTTGCTCTATG	TCCTGACACTTGCTGCTTAC	127	101.797	127
*Aif1*	NM_017196	ATCGTCATCTCCCCACCTAA	GATCATCGAGGAAGTGCTTGT	145	92.694	145
*Bdnf*	NM_001270630.1	GGTCGATTAGGTGGCTTCATAG	CGGAAACAGAACGAACAGAAAC	160	98.044	160
*Cat*	NM_012520	TCCCAACTACTACCCCAACA	AAGTGACGTTGTCTTCATTAGCA	121	103.621	121
*Crhr1*	NM_030999	GGTATACACTGACTACATCTACCAG	CAGCCTTCCTGTACTGAATGG	143	100.945	143
*Fkbp4*	NM_001191863.1	TCATCAAGAGAGAGGGTACAGG	TGGTTGCCACAGCAATATCC	183	103.394	183
*Fkbp5*	NM_001012174	CACCAGTAACAATGAAGAAAACCC	CCTCACTAGTCCCCACTCTT	116	108.288	116
*Foxm1*	NM_031633	GGCTTGGAAAGATGAGTTCTGA	AACCTTAACCCGATTCTGCTC	101	103.778	101
*Gpx1*	NM_030826.4	ACATCAGGAGAATGGCAAGAATG	CATTCACCTCGCACTTCTCAAAC	110	104.432	110
*Gss*	NM_012962	GACAGGAAGATCCATGTAATCCG	ATCTCGGAAGTAAACCACAGC	120	101.884	120
*Hsp90aa1*	NM_175761.2	AAACAGCACTCCTGTCTTCC	GCCTAGTCTACTTCTTCCATGC	199	103.447	199
*Hspa4*	NM_153629.1	ACCACCTCAAGCAAAGAAGG	CCGTTCCTTCTCCAGTTTATCC	154	97.097	154
*Itgam*	NM_012711	GATGTTCAAGCAGAATTTCGGT	GTATTGCCATCAGCGTCCAT	117	106.195	117
*Nfe2l2*	NM_031789	CAGCATGATGGACTTGGAATTG	AGTTGCTCTTGTCTCTCCTTTTC	189	108.400	189
*Nr3c1*	NM_012576.2	CCTTTGTTCTAAGCTAGGGAAGG	GTGGATGAGGATGGTTAGAATGG	127	96.072	127
*Nr3c2*	NM_013131.1	GGCAAATCTCAACAACTCAAGG	TGAAGTGGCATAGCTGAAGG	142	105.798	142
*P2ry12*	NM_022800	TCCGAGTCAACAGAATAACCAG	GATCTTGTAGTCTCTGCTGCAC	113	94.093	113
*Sod1*	NM_017050	GACAATACACAAGGCTGTACCAC	TTGCCCAAGTCATCTTGTTTCTC	230	109.360	230
*Tmem119*	NM_001107155	CGAGACAGTTGGACCGAGAC	ACAAGTAGCAGCAGAGACAGGAG	141	99.560	141

## Data Availability

Data will be available upon request.

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
