# Peer review of "Acute Ozone-Induced Transcriptional Changes in Markers of Oxidative Stress and Glucocorticoid Signaling in the Rat Hippocampus and Hypothalamus Are Sex-Specific"

_ijms, 2023, doi:10.3390/ijms24076404_

Round 1

Reviewer 1 Report

Valdez et al. demonstrated how O3 causes oxidative stress in the brain, particularly in the hippocampus and hypothalamus, and that males and females are affected differently. Interestingly, this study shows some exciting differences between males and females, such as the mitochondrial complex II activity. The author claims that genes related to glucocorticoid signaling show the most sex-specific effects of ozone in the hippocampus and hypothalamus. I have some concerns that need to address. 

1. There are sex-dependent differences (baseline), and there are ozone-induced changes that are statistically different. It is difficult to keep track of these differences to understand what they mean. A summary diagram/table or plot (heatmap) will help readers understand the key findings of this paper. 

2. The author's key conclusion is that ozone exposure affects glucocorticoid signaling in a sex-specific manner, in which only Nr3c1 and Nr3c2 showed the most apparent sex-specific responses to ozone. The other markers with a statistically significant sex-specific reaction include the fkbp5. The statistical significance here was misleading because both females and males showed increased fkbp5 in response to ozone, but only the female reached statistical significance. Given the importance of Nr3c1 and Nr3c2 to the conclusion, authors should discuss the significance of these two markers and speculate why they respond to ozone in a sex-dependent manner.

Reviewer 2 Report

Authors investigate: sex specific Ozone-Induced Transcriptional Changes in Markers of Oxidative Stress and Glucocorticoid Signaling in Rat 

One area for concern is the n value of 4. Is this sufficiently powered?

Is there a range of "good" ozone and "bad" ozone-for example greater than 0.2 ppm for 4 h/day for two consecutive days is bad?

Line 54: which sex was more affected?

Line 63: previous studies, what outcomes were typically measured?

The first paragraph of the results seems better suited for the introduction

Figure 2/3: is n =4 sufficiently powered to do anova analysis?

Line 264+ - the authors should be explicit that the data refers to mRNA

Line 404-how was sex determined at culling?

Line 421-why this concentration of ozone?
